# Neural Transmitted Radiance Fields

**Chengxuan Zhu**[†]
Nat'l Eng. Research Center of Visual Technology
School of Computer Science
Peking University
peterzhu@pku.edu.cn

**Renjie Wan**[†]
Department of Computer Science
Hong Kong Baptist University
renjiewan@comp.hkbu.edu.hk

**Boxin Shi**[*]
Nat'l Eng. Research Center of Visual Technology
School of Computer Science
Peking University
shiboxin@pku.edu.cn

## Abstract

Neural radiance fields (NeRF) have brought tremendous progress to novel view synthesis. Though NeRF enables the rendering of subtle details in a scene by learning from a dense set of images, it also reconstructs the undesired reflections when we capture images through glass. As a commonly observed interference, the reflection would undermine the visibility of the desired transmitted scene behind glass by occluding the transmitted light rays. In this paper, we aim at addressing the problem of rendering novel transmitted views given a set of reflection-corrupted images. By introducing the transmission encoder and recurring edge constraints as guidance, our neural transmitted radiance fields can resist such reflection interference during rendering and reconstruct high-fidelity results even under sparse views. The proposed method achieves superior performance from the experiments on a newly collected dataset compared with state-of-the-art methods. Our code and data is available at https://github.com/FreeButUselessSoul/TNeRF.

## 1 Introduction

Novel view synthesis plays a vital role in various computer vision applications. Tremendous progress has been witnessed recently with the introduction of neural radiance fields (NeRF) [1]. By modeling the emitted radiance and density of a scene from a set of images with a multi-layer perceptron (MLP), it demonstrates an unprecedented level of fidelity on a range of challenging scenes. Despite its success in diverse scenarios, such a framework still faces challenges when working with glass, a medium which is frequently observed in photography and daily life [2] to provide see-through protection for desired scenes. However, glass's reflectivity may reflect undesired scenes on images captured through it. People always feel interested in the transmitted scenes behind the glass and want to remove the undesired reflection. This traps neural radiance fields into a dilemma: Its well-designed framework is capable of modeling every detail in a scene, including those *undesired* ones.

One plausible way to resolve this dilemma is opting for a pre-processing tool to remove the undesired components before the captured images are fed into NeRF [1]. For example, existing reflection removal methods [3, 4, 5] can be adopted to suppress the undesired components and provide better visibility for transmitted scenes. However, the processed images from different views may violate

---

[†]Equal Contribution.
[*]Corresponding author.

36th Conference on Neural Information Processing Systems (NeurIPS 2022).

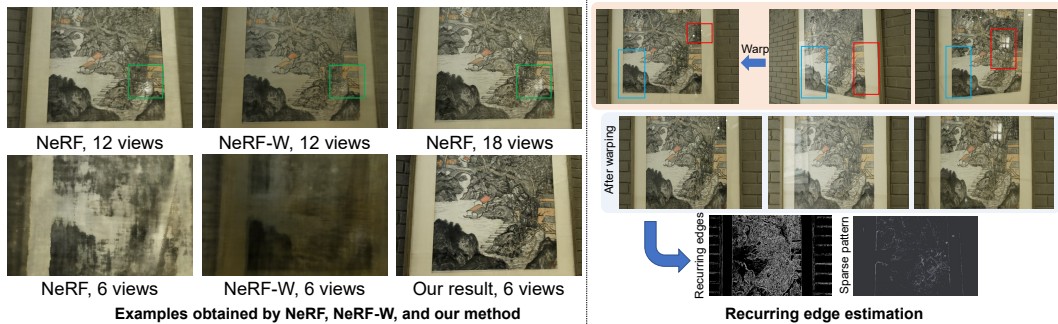

Figure 1: Left: The results obtained by NeRF [1] and NeRF-W [7] with 6 and 12 training views, the result obtained by NeRF [1] with 18 training views, and our result. The rendered results are with low reconstruction fidelity for NeRF [1] and NeRF-W [7] only with 6 and 12 training views. For NeRF [1] with 18 training views, the result shows higher fidelity, but the undesired reflection is also finally rendered (labeled by green box). Right: The recurring edge estimation process. By warping images from neighboring views into a predefined view, the recurring edges belonging to the transmission can be extracted to guide the novel transmitted view synthesis. The blue and red boxes denote the repeated transmission and sparse reflection pattern, respectively.

the assumption of NeRF that the scene is photometrically static. Besides, the reflection cannot be completely removed in most cases by existing single-image-based reflection removal approaches [6], and the remaining reflection may be modeled by NeRF as a part of the scenes, undermining the quality of rendered results.

An alternative solution is to render with undesired components like NeRF-W [7] by separating the whole scene into static and transient elements with exclusive properties. However, the reflection and transmission usually present a highly "twisted" relationship since they may influence each other during light transportation and reflect such ill-posed essence in captured images. From results shown in Figure 1(left), it may challenge the applicability of a purely unsupervised strategy in NeRF-W [7], thus showing degraded performance. Besides, NeRF-W [7] also partially relies on a *dense* set of multi-view images to conduct the separation. As an impractical demand inherited from NeRF [1], it contradicts the casual photographing experience, where only limited inputs with sparse views are available. The results in Figure 1(left) illustrate that NeRF-W [7] cannot perform high-fidelity synthesis with limited views, let alone effective separation between the transmitted scene and the reflected scene. Thus, how to render *novel transmitted scenes* with *sparse views* poses unique challenges.

In contrast to the exclusive properties required by NeRF-W [7], we employ the recurring edges, a phenomenon verified by pioneering reflection removal methods [8, 9], to link the above issues by warping images with different viewpoints into a predefined target view. As shown in Figure 1 (right), given an aligned image sequence, the transmitted scenes appear in different views with a large overlapping area while the reflected scenes only have a sparse presence [8]. Besides, the aligned image features from the neighboring views can also be used for a more high-fidelity reconstruction of the target view. We, therefore, design a method supported by two pillars: 1) Edges of the repeated patterns, denoted as the recurring edges shown in Figure 1(right), that could be estimated as a pilot to guide the differentiation during rendering; 2) features from reference views, which are incorporated into rendering for high-fidelity reconstruction under sparse views.

Several modifications have to be conducted to accommodate the above designs. Since the transmission/reflection edges only show their significance in a certain area, the classical "pixel-wise" rendering framework cannot capture such internal physical properties. We employ a patch-based rendering framework, which aggregates the pixels from a certain area into a patch as the training batch. Besides, due to the interference caused by the reflection, the features extracted by a network pre-trained for general tasks [10] may introduce interference from feature level, which deteriorates the transmission/reflection separation described before. We, therefore, propose a transmission encoder to penetrate the feature-level interference caused by the reflection and render the transmission structure more accurately. At last, since the strength of reflection tends to rely on the viewing direction,

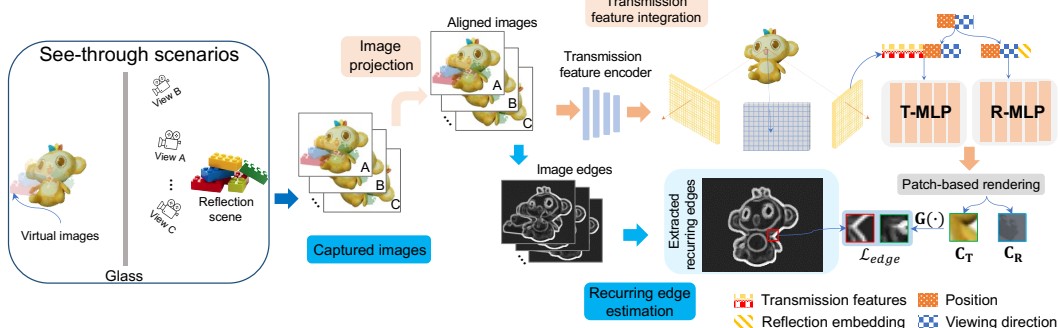

Figure 2: The proposed framework. Given images captured through glass from several different views, our method first projects the images at neighboring views to their target view (image projection). Next, a transmission feature encoder extracts the transmission features from the reference images (transmission feature integration). The extracted transmission features are then passed into the transmission MLP (T-MLP) for transmission rendering ($\mathbf{C_T}$). For the reflection rendering ($\mathbf{C_R}$), we feed the viewing direction and corresponding reflection embeddings into the reflection MLP (R-MLP). The recurring edges repeatedly appearing in the aligned image sequence are extracted to further differentiate the transmission and reflection based on the edge loss $\mathcal{L}_{edge}$ (recurring edges estimation).

the weighting coefficients for the two layers are obtained from the viewing direction to combine transmission and reflection layers.

In summary, the above considerations lead to our Neural Transmitted Radiance Fields as depicted in Figure 2, and its main contribution can be concluded as follows:

- An approach to render novel transmitted views, which is capable of penetrating the reflection interference.
- A unified consideration derived from recurring edges for removing reflection/transmission ambiguity and increasing rendering fidelity of transmission.
- A dataset collected from the real world for evaluation.

## 2 Related work

**Novel view synthesis with NeRF.** Novel view synthesis aims at generating images from another viewpoint using a set of established views. The classical approaches rely on the explicit representations like multi-plane images [11, 12], meshes [13], or point cloud [14] to render novel views. Despite their rendering efficiency at test time, they usually suffer from the limited expressiveness within their representation. In contrast to the explicit representations, Neural Radiance Fields (NeRF) [1] utilize the implicit representation to represent 3D scenes using neural networks. For example, NeRF [1] approximates a continuous 3D MLP by mapping an input 5D coordinate to its corresponding scene properties. Given its promising representation ability, NeRF [1] has been extended to solving problems like relighting and scene editing. Recently, NeRF-W [7] proposes to adapt the original NeRF [1] to wild scenes by introducing an additional MLP to model the undesired transient element, which grants the original NeRF [1] extended ability to model the scenes with photometric variations and transient components. Since NeRF requires a large number of images for training, several methods [10, 15] are proposed to make it feasible with sparse views by utilizing the feature information from neighboring views. In this work, we aim to extend NeRF to see-through scenarios by eliminating negative interference caused by reflection.

**Transmission recovery from reflection-corrupted images.** How to recover the transmitted scene from reflection-corrupted images has been discussed for decades. Early methods [16, 17] make use of the edge difference to reduce the ambiguity between the transmission and reflection through optimization. Recent methods mainly adopt the deep learning framework to solve this problem. For example, Fan *et al.* [18] proposed a two-stage deep learning approach to learn the mapping between

the mixture images and the estimated clean images. Wan *et al.* [19, 20] proposed a cooperative model to better preserve the background details. Zhang *et al.* [21] proposed a perceptual reflection removal method based on the generative adversarial network. Wei *et al.* [5] proposed another method to solve the reflection removal problem with non-aligned image pairs. Besides, some recently proposed [4] methods also try to focus on the reflection-corrupted regions by localizing the reflection first. The recently proposed method [22] also removed reflections under the guidance of an additional polarizer. Instead of solely relying on single images, plenty of methods [8, 23, 24, 25] also leverage advantages from the priors provided by multiple images to remove reflections based on recurring edges belonging to the transmission. Such inconsistency observed from different views also allows our framework to differentiate transmission and reflection during rendering.

## 3   Preliminaries about Neural Radiance Fields

We first briefly review the settings in NeRF [1] and NeRF-W [7]. A NeRF represents a scene as a continuous volumetric radiance field $f$, which maps the position of any given point $\mathbf{x} = (x, y, z)$ and a viewing direction $\mathbf{d} = (\theta, \phi)$ into a color $\mathbf{c} = (r, g, b)$ and density $\sigma$ as $f(\mathbf{x}, \mathbf{d}) = (\mathbf{c}, \sigma)$. Then, the expected color $\hat{\mathbf{C}}$ corresponding to the camera ray $\mathbf{r}(t) = \mathbf{o} + t\mathbf{d}$ emitted from the center of projection of a camera at position $\mathbf{o}$ can be obtained as

$$\hat{\mathbf{C}}(\mathbf{r}) = \int_{t_n}^{t_f} T(t)\sigma(t)\mathbf{c}(t)dt, \tag{1}$$

where $t_n$ and $t_f$ denote the near and far bounds of rendering, and $T(t) = \exp(-\int_{t_n}^{t} \sigma(\mathbf{r}(s))ds)$ denotes the accumulated transmittance. NeRF-W [7] extends NeRF [1] with two MLPS to model the desired static ($f_s$) and undesired transient ($f_t$) components as

$$f_s(\mathbf{x}, \mathbf{d}, \ell_a) = (\mathbf{c}_s, \sigma_s) \text{ and } f_t(\mathbf{x}, \mathbf{d}, \ell_t) = (\mathbf{c}_t, \sigma_t). \tag{2}$$

Due to the introduction of the additional MLP for transient elements, as well as the embeddings $\ell_a$ and $\ell_t$ adapting to the various appearances and transient objects, NeRF-W [7] is capable of decomposing the desired static components during rendering.

For an image $\mathbf{I}$ captured in see-through scenarios, the relationship between the reflection $\mathbf{R}$ and transmission $\mathbf{T}$ can be formulated as $\mathbf{I} = \beta\mathbf{T} + \alpha\mathbf{R}$ [6], where $\beta$ and $\alpha$ denotes the weighting coefficients. Applying NeRF-W [7] to achieve novel transmitted view synthesis is not directly applicable due to the following issues: 1) Dense view requirement: Both NeRF-W [7] and NeRF [1] inherit the substantial demand for dense multi-view images, and their performances degenerate under real-world scenarios where only limited input with sparse views are available. 2) Undesired reflection interference: For the objects on both sides of the glass, their emitted light rays may be absorbed, refracted, and reflected, leading the light energy attenuation or highly mixed phenomenon in the captured image [6]. This makes their separation present a ill-posed essence, which cannot be simply resolved by using an unsupervised model. We propose to construct Neural Transmitted Radiance Fields to address these issues.

## 4   Neural Transmitted Radiance Fields

Our goal is to learn radiance fields as a representation for the transmitted scene behind glass. To avoid the challenging separation ambiguity for transmission/reflection and low reconstruction fidelity under sparse views, we build a unified approach by warping pixels to the desired target views. By extracting the recurring edges after warping, our approach can get rid of the interference from the reflection. Meanwhile, a transmission feature integration scheme is introduced to merge the transmission features from the warped images, which enables higher reconstruction fidelity even under sparse views.

The pixel-wise rendering scheme in NeRF [1] and NeRF-W [7] lacks the ability to reflect the transmission/reflection correlation, since some statistical properties can only be observed in a certain area. To better consider the spatial correlation among neighboring pixels, we employ the patch-based rendering scheme. Specifically, during training we sample a $\mathcal{K} \times \mathcal{K}$ patch $\mathcal{P}(\mathbf{u}, s)$ in a given image following the way proposed in [26] as follows:

$$\mathcal{P}(\mathbf{u}, s) = \left\{ (sx + u, sy + v) \mid x, y \in \left\{ -\frac{\mathcal{K}}{2}, \ldots, \frac{\mathcal{K}}{2} - 1 \right\} \right\}, \tag{3}$$

where $\mathbf{u} = (u, v)$ denotes the center position of the patch, $x$ and $y$ range from $-\frac{\mathcal{K}}{2}$ to $\frac{\mathcal{K}}{2} - 1$ to create a sampling grid, and $s$ denotes the scale of the image patch $\mathcal{P}$. In our setting, we apply a combined sampling of grid sampling with $s = 1$ over the entire image and random sampling with $s > 1$, allowing image patches to capture spatial information of different scales. The aggregated pixels from the image patch are then used for training, during which the model can not only learn the colors and densities through one ray, but also get a thorough understanding towards the context.

## 4.1 Transmission feature integration

In contrast to NeRF [1] and NeRF-W [7], which solely rely on point coordinates and viewing direction as the input, we propose to address the fidelity issue by means of the additional guidance from reference images. Given a scene only with sparse views, the feature guidance from neighboring views can help to reconstruct novel target views with higher fidelity. In general, such image features are extracted by using a 2D CNN network [10] pretrained for general tasks. However, since the images under neighboring views are also captured through glass, a general CNN network for view synthesis like previous methods [15] may lead to feature-level interference during rendering.

From Figure 2, we first project the neighboring views to their target view. To resist such interference during feature integration, we propose to obtain the transmission features by means of a transmission encoder $\eta$ obtained by pre-training a reflection removal network, *e.g.*, ERRNet [5]. As a feature encoder pretrained for reflection removal task, $\eta$ aims at extracting the transmission features as $\mathbf{W} = \eta(\mathbf{I})$, where $\mathbf{W}$ denotes the extracted feature volume. Instead of solely relying on the single-scale feature like pixel-NeRF [27], which suffers from view misalignment in our setting, we propose to build a feature pyramid as $\mathbf{W} = \{\mathbf{W}_g, \mathbf{W}_l\}$, where $\mathbf{W}_g$ and $\mathbf{W}_l$ denote the global and local features with different scales, respectively. In general, as CNNs naturally store local information in shallow layers and encode global information in deeper layers with a large receptive field, we extract $\mathbf{W}_g$ and $\mathbf{W}_l$ from the bottleneck layer and second last layer of $\eta$, respectively. Prior works also suggest combining features from different scales can lead to better convergence and geometric details [28]. The two feature maps are upsampled using Nearest-Neighbor Interpolation to the size of the image.

The extracted global and local transmission features for the $i$-th target view are then passed into the MLP network for the transmission rendering (T-MLP) $f_{\mathbf{T}}$, along with the position and viewing direction, as

$$f_{\mathbf{T}}(\mathbf{x}, \mathbf{d}, \mathbf{W}_g^{(i)}, \mathbf{W}_l^{(i)}) = (\mathbf{c}_t^{(i)}, \sigma_t^{(i)}), \tag{4}$$

where $c_t^{(i)}$ and $\sigma_t^{(i)}$ are the estimated color and density of the transmitted scene for the $i$-th view. As for reflection, a learnable reflection embedding $\ell_r^{(i)}$ is introduced for each view $i$, and the color $c_r^{(i)}$ and density $\sigma_r^{(i)}$ of the reflection scene are estimated using an additional MLP network for the reflection rendering (R-MLP) as

$$f_{\mathbf{R}}(\mathbf{x}, \mathbf{d}, \ell_r^{(i)}) = (\mathbf{c}_r^{(i)}, \sigma_r^{(i)}), \tag{5}$$

where the reflection embedding is optimized along with the weights of $f_{\mathbf{R}}$. Similar to existing NeRFs [1, 7], we construct a self-consistent loss as follows:

$$\mathcal{L}_{sc} = \sum_{p(\mathbf{r}) \in \mathcal{P}} \|\hat{\mathbf{C}}(\mathbf{r}) - \mathbf{C}(\mathbf{r})\|_1, \tag{6}$$

where $p(\cdot)$ denotes the mapping from a ray to its corresponding pixel position, and $\hat{\mathbf{C}}(\mathbf{r})$ denotes the observed color and $\mathbf{C}$ denotes the composite of $\mathbf{C_T}$ and $\mathbf{C_R}$ physically defined as $\mathbf{C} = \beta\mathbf{C_T} + \alpha\mathbf{C_R}$. Here $\alpha$ denotes the weighting coefficients to balance $\mathbf{C_T}$ and $\mathbf{C_R}$. For simplicity, we directly set $\beta = 1 - \alpha$ like previous reflection removal methods [29].

Several factors during light transportation can influence the weighting coefficients $\alpha$. In our setting, since the strength of reflection largely depends on viewing direction [30] rather than the exact camera position, we propose to model such influence by a simple MLP that outputs the weighting coefficients given corresponding viewing direction encoding as $f_\alpha(\mathbf{x}, \mathbf{d}) = \alpha$. Specifically, as $\mathbf{d}$ and $\mathbf{x}$ refer to the viewing direction and the position of any given points, respectively, the weighting map of a given view is rendered similarly to Equation (1) by considering the two factors. It enables the network to increase its robustness for general cases.

## 4.2  Recurring edge constraints

The settings described above ensure a high-fidelity novel view reconstruction even under sparse views. However, since no reflection removal model can reach a completely reflection-free status, the reflection residuals left by the transmission feature encoder may further deteriorate the ambiguity issue. We then utilize the benefits provided by the aligned image sequence to further disentangle their twisted relationship.

As pointed by pioneer methods [9, 29], when multiple images with different viewpoints are captured through glass, the transmission appears frequently across all samples, while the reflection may only have sparse presence. This property makes the recurring transmission edges become a "pilot" to relieve ambiguity between the transmission and reflection. In our method, we first transform aligned images into their corresponding edge domain and measure the frequency of edge occurrence as

$$\phi(\mathbf{z}) = \frac{\sum_{i=1}^{k} G_i(\mathbf{z})^2}{(\sum_{i=1}^{k} G_i(\mathbf{z}))^2}, \tag{7}$$

where $G_i(\mathbf{z})$ denotes the gradient magnitude at pixel location $\mathbf{z}$ of view $i$, and $k$ denotes the total number of pixels. It measures the sparsity of the vector which achieves its maximum value of $1$ (when only one non-zero item exists) and achieves its minimum value of $1/k$ (when all items are non-zero and have identical values). Using this measurement, each edge pixel is assigned with two probabilities regarding whether it belongs to the transmitted scene or not.

Recognizing the transmission and reflection poses a great challenge for a vanilla NeRF network aiming at multi-view consistency, we provide our network with a likelihood estimation of the transmission and reflection, following the strategy proposed in [29]:

$$l_{\mathbf{T}_i}(\mathbf{z}) = s(-(\phi(\mathbf{z}) - \frac{1}{k})),$$
$$l_{\mathbf{R}_i}(\mathbf{z}) = s(\phi(\mathbf{z}) - \frac{1}{k}), \tag{8}$$

where $s$ is a sigmoid function to facilitate the separation. Then, the final likelihood for transmission edges can be defined as follows:

$$E_{\mathbf{T}}(\mathbf{z}) = \begin{cases} 1, & l_{\mathbf{T}_i/\mathbf{R}_i}(\mathbf{z}) > 0.6 \\ 0, & \text{otherwise.} \end{cases} \tag{9}$$

From the results shown in Figure 1(right), the likelihood of target transmitted scene effectively labels the transmission edges, which provides a clear guidance during rendering. We search from 0 to 1 with a step of 0.1 and fix the threshold in Equation (9) as 0.6 in our experiments.

Based on the binarized edge map obtained in Equation (9), we propose to constrain rendering using a simple edge loss as follows:

$$\mathcal{L}_{edge} = \sum_{p(\mathbf{r}) \in \mathcal{P}} \|E_{\mathbf{T}}(p(\mathbf{r})) \odot (\hat{\mathbf{C}}(\mathbf{r}) - \mathbf{C}(\mathbf{r}))\|_2^2, \tag{10}$$

which means we constrain the rendered image more on wherever the transmission edges are present, encouraging a more faithful reconstruction towards the mixture image.

Besides, the transmission edges and those of the reflection tend to be different, as prior works pointed out [21]. We inherit the exclusion assumption between the gradient of transmission and reflection to differentiate the two parts in the gradient domain as follows:

$$\mathcal{L}_{\text{excl}}(\theta) = \|\Psi(\mathbf{C_T}, \mathbf{C_R})\|_F,$$
$$\Psi(\mathbf{C_T}, \mathbf{C_R}) = \tanh(\lambda_{\mathbf{T}}|G(\mathbf{C_T})|) \odot \tanh(\lambda_{\mathbf{R}}|G(\mathbf{C_R})|), \tag{11}$$

where $\lambda_{\mathbf{T}} = \sqrt{\frac{\|G(\mathbf{C_R})\|_F}{\|G(\mathbf{C_T})\|_F}}$ and $\lambda_{\mathbf{R}} = \sqrt{\frac{\|G(\mathbf{C_T})\|_F}{\|G(\mathbf{C_R})\|_F}}$. The downsampling operation in its original setting is discarded since our patch has been small enough. $\Psi(\cdot, \cdot)$ defines a pixel-wise correlation between the transmission and reflection, which helps to separate them in the gradient domain.

By considering the above settings and constraints, our loss functions for the whole optimization can be concluded as $\mathcal{L} = \gamma \mathcal{L}_{sc} + \delta \mathcal{L}_{edge} + \omega \mathcal{L}_{excl}$, where $\gamma = 2$, $\delta = 0.002$, and $\omega = 1$ are the weighting coefficients to balance the influence of each term. All the calculations and optimization processes are based on the patch sampled from the image.

### 4.3 Implementation details

We implement our framework using PyTorch. In the training and testing phase, two eight-layer MLPs with $256$ channels are used to predict colors $\mathbf{c}$ and densities $\sigma$ corresponding to the transmitted and reflection scenes. We train a "coarse" network along with a "fine" network network for importance sampling. We sample 64 points along each ray in the coarse model and 64 points in the fine model. A batch contains an image patch of $32{\times}32$ pixels, equivalent to 1024 rays. Similar to the settings in NeRF [1], positional encoding is applied to input location before they are passed into the MLPs. We use the Adam optimizer with defaults values $\beta_1 = 0.999$, $\beta_2 = 0.9$, $\epsilon = 10^{-8}$, and a learning rate $10^{-4}$ that decays following the cosine scheduler during the optimization. We optimize a single model for about 100K iterations on two NVIDIA V100 GPUs.

Our transmission encoder can extract transmission features of multiple resolutions by leveraging advantages from ERRNet [5] and U-Net [31]. Based on our experiments, even a simple U-Net [31] can also help to resist the reflection interference during rendering. The transmission encoder is first integrated with a feature decoder to compress the information to a lower dimension, and the output is fed into the transmission MLP. For the pre-training of the transmission feature encoder, we follow the strategy proposed in [5] with its release training data. Thus, the model is approximately equivalent to its release model.

## 5 Experiments

**Dataset.** Our experiments are based on a real-world dataset we collect. This dataset contains $8$ different real-world scenes, each consisting of 20 to 30 mixture images with different poses. Specifically, $4$ scenes are with the ground truth for quantitative evaluations in the experiments. We follow the setup proposed in [6] by first capturing the mixture image through transparent glass and then its corresponding transmitted scene by removing the glass. We follow the previous settings in NeRF [1] and its variants [10, 32] to estimate the pose via COLMAP [33, 34]. Like most reflection removal methods [6], we assume a piece of planar glass when capturing images. Though some glasses in the real world are slightly curved, it does not obviously affect the robustness of our method. We also test our network on the LLFF dataset [35] and the RFFR dataset [32].

**Baselines.** We compare our method with five NeRF-based methods: 1) NeRF [1]: the original NeRF method; 2) NeRF-W [7]: an unofficial implementation of NeRF in the Wild*; 3) MVSNeRF [10]: a NeRF-based multi-view stereo model for novel view synthesis (NVS) which also relies on reference views; 4) Reflection removal (RR) + NeRF: the most direct way for novel transmitted view synthesis, namely training a NeRF network based on the results of a reflection removal method (without losing generality, we use ERRNet [5], a state-of-the-art method inspiring our design); 5) NeRFReN [32]: a NeRF-based method designed for see-through scenarios released recently; 6) RR + MVSNeRF: NVS using MVSNeRF [10] with reflection removal applied as a pre-processing. All the results in Section 5 are obtained using six views for training. Due to the page size limitation, the results of other methods trained under more views can be found in the supplementary material.

**Metrics.** We report quantitative performance using PSNR, SSIM and LPIPS [36]. For PSNR and SSIM, higher value indicates better performance. For LPIPS [36], lower value indicates better performance. By measuring image fidelity using high-level features, LPIPS [36] can better match human judgements of image similarity.

### 5.1 Qualitative results

The qualitative results of the rendered novel transmitted views are presented in Figure 3 and Figure 4. Due to the effectiveness of the proposed transmission feature integration scheme, our method can reliably reconstruct novel transmitted views under only six views. Besides, the feature integration scheme and its cooperative recurring edge constraints further reduce the ambiguity between reflection and transmission during rendering. For other methods, though NeRF [1] can partially reconstruct the desired transmitted views, it still suffers from the interference caused by the reflection; NeRF-W [7] and NeRFReN [32] output degenerated results as they cannot separate reflection from transmission

---

*https://github.com/kwea123/nerf_pl

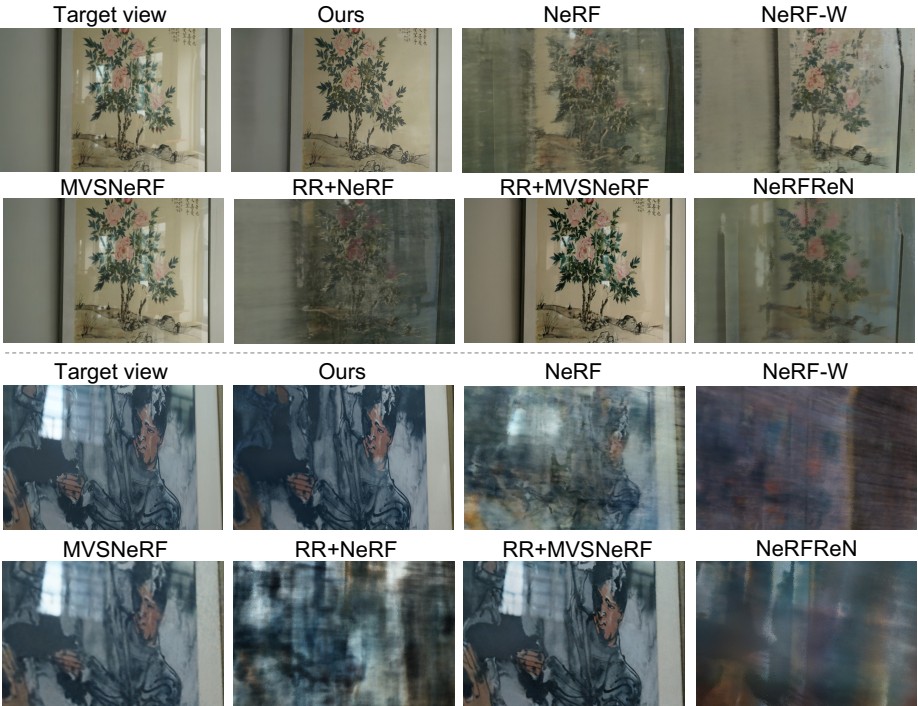

Figure 3: From left to right: the target view, our results, the results obtained by NeRF [1], NeRF-W [7], MVSNeRF [10], RR+NeRF, RR+MVSNeRF [10], and NeRFReN [32]. The target views in the two scenarios (scenes with non-detachable glass, *e.g.*, painting with fixed frames) are captured through glass for reference only (no ground truth transmission). Please zoom in for more details.

with as few inputs as our method; MVSNeRF [10] can keep high fidelity of the rendering results like our methods, but obvious reflection residuals still remain in their results since their feature extractor cannot resist the reflection interference.

Table 1: Quantitative evaluation compared with NeRF [1], NeRF-W [7], MVSNeRF [10], RR+NeRF, NeRFReN [32], and RR+MVSNeRF [10]. Higher PSNR and SSIM values denote better results (↑), while lower LPIPS values denote better results (↓).

|  | PSNR ↑ | SSIM ↑ | LPIPS ↓ |
|---|---|---|---|
| Ours | **22.75** | **0.841** | **0.205** |
| NeRF [1] | 16.72 | 0.644 | 0.510 |
| NeRF-W [7] | 15.44 | 0.624 | 0.529 |
| MVSNeRF [10] | 17.83 | 0.685 | 0.400 |
| RR+NeRF [1] | 18.42 | 0.691 | 0.539 |
| NeRFReN [32] | 18.63 | 0.716 | 0.449 |
| RR+MVSNeRF [10] | 18.16 | 0.679 | 0.447 |

## 5.2 Quantitative results

The quantitative results of the rendered novel transmitted view in Table 1 also validate the observation in Figure 4. Higher PSNR values show that our method can render novel transmitted views and recover the color information with higher accuracy. Higher SSIM values indicate that our method can preserve the structural information with high-frequency details. Lower LPIPS values show that recovered images by our method better aligns with human perception. Specifically, for the results obtained by other methods, their higher LPIPS values indicate that they cannot reconstruct realistic results with high fidelity when only six views are provided. Specifically, MVSNeRF [10] achieves the second best LPIPS results among all methods, which shows that the feature integration scheme

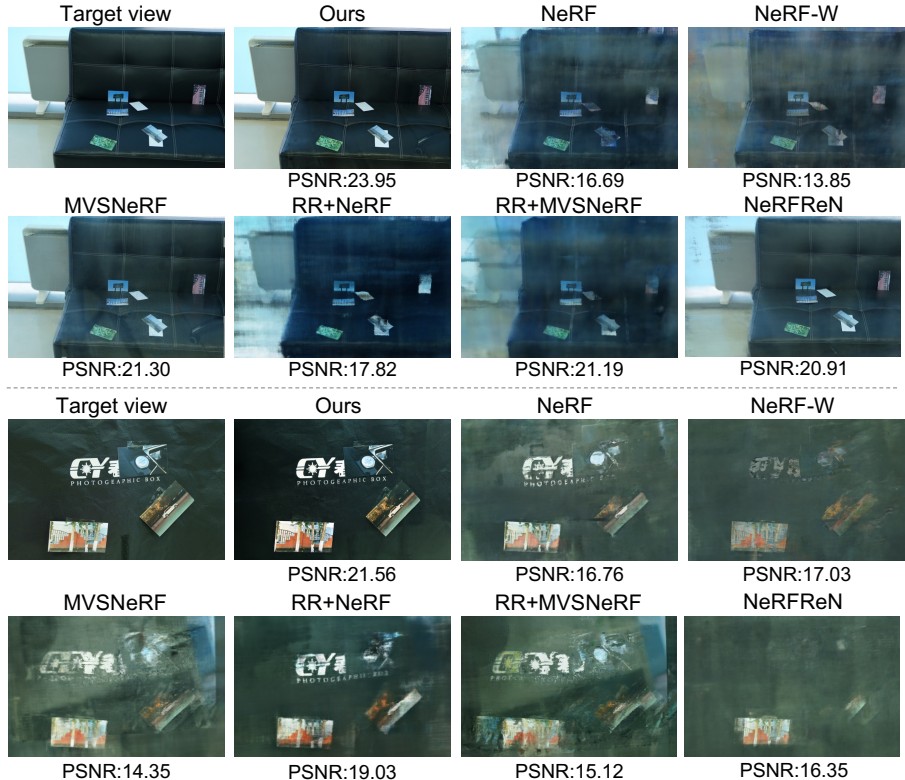

Figure 4: From left to right: the target view, our results, the results obtained by NeRF [1], NeRF-W [7], MVSNeRF [10], RR+NeRF [1], RR+MVSNeRF [10], and NeRFReN [32]. The target views in the two scenarios (scenes with detachable glass, captured with the similar setup as [6]) are obtained by removing the glass, and are compared with the results as ground truth for PSNR values shown below each image. Please zoom in for more details.

Table 2: Ablation study on the model without the transmission encoder ($\eta$), the recurring edge constraints (REC), the patch-based rendering scheme ($\mathcal{P}$), and either the transmission encoder or patch-based rendering scheme. Higher PSNR and SSIM values denote better results (↑), while lower LPIPS values denote better results (↓).

|  | PSNR ↑ | SSIM ↑ | LPIPS ↓ |
|---|---|---|---|
| Complete model | **22.75** | **0.841** | **0.205** |
| (w/o) $\eta$ | 15.60 | 0.512 | 0.565 |
| (w/o) REC | 22.48 | 0.836 | 0.265 |
| (w/o) $\mathcal{P}$ | 18.17 | 0.769 | 0.271 |
| (w/o) $\eta$ & $\mathcal{P}$ | 15.65 | 0.588 | 0.594 |
| NeRF | 16.72 | 0.644 | 0.510 |

can indeed improve the rendering fidelity. On the other hand, RR+NeRF setting achieves the second best PSNR and SSIM result, which shows the effectiveness of this simple strategy to some degrees.

## 5.3 Ablation study

Our network consists of two parts: the transmission encoder to disentangle the transmitted and reflected scenes, and the patch-rendering scheme to utilize the physical constraints. We conduct several experiments to evaluate the benefits of these two parts. We first remove the transmission encoder and directly feed the MLP with the position and viewing direction. From the results shown in Figure 5, without the support of reference views, the position and viewing directions cannot effectively reconstruct the desired transmitted views. The errors shown in Table 2 also become closer

| Target view | Full model | W/o REC | W/o $\mathcal{P}$ | W/o $\eta$ | W/o $\eta$ & $\mathcal{P}$ |

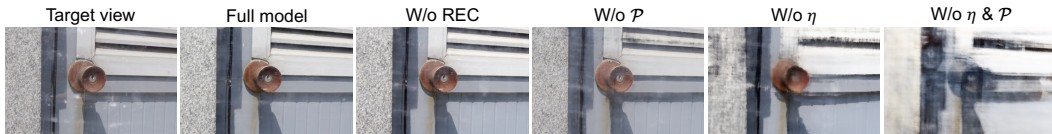

Figure 5: From left to right: the target view (with reflection), the results of our complete model, without the recurring edge constraints (REC), without the transmission encoder ($\eta$), without patch-based rendering scheme ($\mathcal{P}$), and with neither transmission encoder nor patch-based rendering scheme.

(a) MVSNeRF, 6 training views, mean PSNR: 18.09    (b) Ours, 6 training views, mean PSNR: **21.69**

Figure 6: Non-reflective scenes from LLFF dataset [35]. Please use Adobe Acrobat to see the animated results.

to those of NeRF in Table 1. We then remove the recurring edge constraints, and more reflection is observed in the rendered target view since the network attaches equal importance to every pixel in the patch. We further remove the patch-based rendering scheme entirely. In this situation, reflection evidently remains on the generated target view due to a lack of physical constraint, as we discussed before. At last, both the patch-based rendering scheme and the transmission encoder are removed, and we achieve a degenerated result similar to that of NeRF-W [7] shown in Table 1. Without loss of generality, we specifically compare with MVSNeRF [10] on non-reflective common examples to show that our method can also adapt to more general scenarios in Figure 6.

## 6 Conclusions

We solve the problem of novel view synthesis for see-through scenarios in this paper. We introduce a transmission encoder to address the fidelity issue caused by sparse views and ambiguity issues led by the reflection interference in a unified framework. Specifically, to further disentangle the twisted relationship between transmission and reflection during the rendering process, we introduce a recurring edge constraint by counting the frequency of edge occurrence among the aligned image sequence. Experimental evaluation on a newly collected dataset demonstrates the promising performance for novel transmitted view synthesis our method could achieve.

**Limitations and future work.** Our method still faces the challenge led by the occlusion issue, since we rely on features from neighboring views to complement the missing information given a sparse set of input views. However, when features are not consistent with a certain view, occlusions may negatively undermine the performance of our proposed method, due to the inaccurate feature information for synthesizing novel views in that area. Besides, our method relies on COLMAP [33, 34] for the rendering, while it may fail on low-transmitted reflections covering large areas or the transmission distortion caused by irregular glass. In this situation, COLMAP [33, 34] cannot correctly extract the transmission features. Investigating NeRF without the requirement for established poses like [37] may help to alleviate this issue. We will also consider the irregular glass for its influence on the transmitted objects' light in our future work.

**Acknowledgement:** This work is supported by National Natural Science Foundation of China under Grant No. 62136001, 61872012. Renjie Wan is supported by the Blue Sky Research Fund under the Research Committee of Hong Kong Baptist University under the Project Number BSRF/21-22/16.

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
