# Supplementary Material:
# Neural Transmitted Radiance Fields

**Chengxuan Zhu**[†]
Nat'l Eng. Research Center of Visual Technology
School of Computer Science
Peking University
peterzhu@pku.edu.cn

**Renjie Wan**[†]
Department of Computer Science
Hong Kong Baptist University
renjiewan@comp.hkbu.edu.hk

**Boxin Shi**[*]
Nat'l Eng. Research Center of Visual Technology
School of Computer Science
Peking University
shiboxin@pku.edu.cn

## A  Clarification for recurring edge estimation

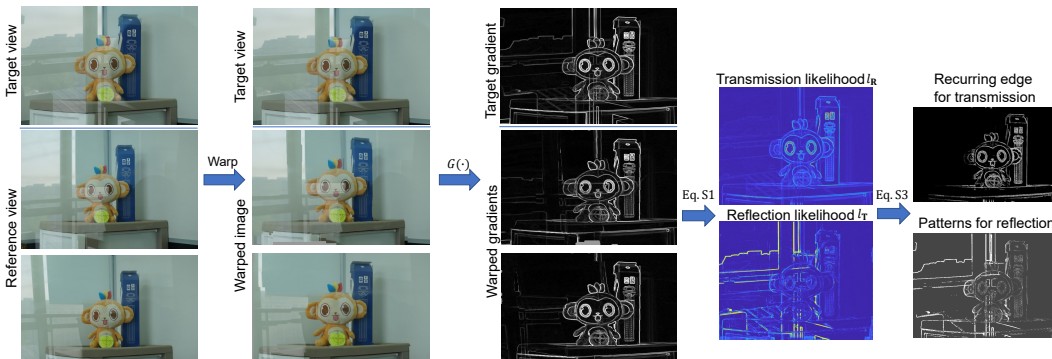

Figure S1: An example for the recurring edge estimation during the training stage. The transmission likelihood and reflection likelihood are displayed with scaled color.

The recurring edges in our framework act as a pilot to guide the training. This is based on a simple observation verified by a number of pioneer reflection removal methods [1, 2]. In this observation, the transmission repeatedly appears among a set of images, while the reflection only has sparse presence. Besides the examples shown in Figure 1 of our main paper, we additionally show an example in this supplementary. From the examples shown in Figure S1, our method first warps images from reference views to a predefined target view. With the warped image sequence, we find the transmission ($l_{\mathbf{T}}$) and reflection ($l_{\mathbf{R}}$) likelihood as follows:

$$
\begin{aligned}
l_{\mathbf{T}_i}(\mathbf{z}) &= s(-(\phi(\mathbf{z}) - \frac{1}{k})), \\
l_{\mathbf{R}_i}(\mathbf{z}) &= s(\phi(\mathbf{z}) - \frac{1}{k}),
\end{aligned}
\tag{S1}
$$

---

[†]Equal Contribution.
[*]Corresponding author.

36th Conference on Neural Information Processing Systems (NeurIPS 2022).

where $\phi(z)$ is defined as

$$\phi(\mathbf{z}) = \frac{\sum_{i=1}^{k} G_i(\mathbf{z})^2}{\left(\sum_{i=1}^{k} G_i(\mathbf{z})\right)^2}. \tag{S2}$$

Equation (S2) measures the sparsity of the vector which achieves its maximum value of 1 (when only one non-zero item exists) and achieves its minimum value of $1/k$ (when all items are non-zero and have identical values). Then, we can further select the desired recurring transmission edges as follows:

$$E_{\mathbf{T}}(\mathbf{z}) = \begin{cases} 1, & l_{\mathbf{T}_i/\mathbf{R}_i}(\mathbf{z}) > 0.6 \\ 0, & \text{otherwise}. \end{cases} \tag{S3}$$

From Figure S1, due to the twisted property between the transmission and reflection, some transmission edges are left in the patterns for reflections. However, this setup has effectively extracted the recurring transmission edges for the follow-up training.

The number of reference images is flexible in our setting. For simplicity, we mainly set this number to 1 or 2 in our experiments.

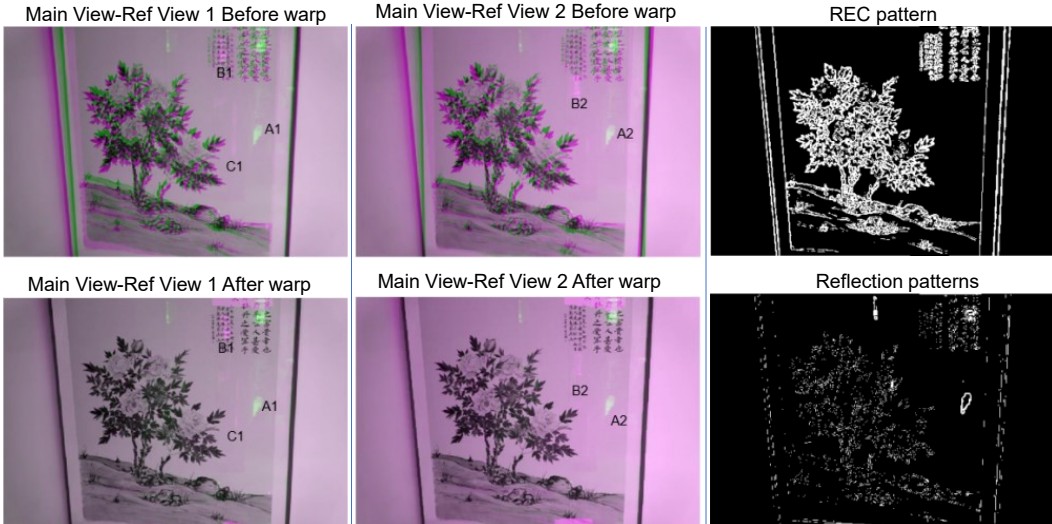

Figure S2: More details about how to the REC extraction when the number of reference views equal to 2.

We further show an example with two reference views in Figure S3. We use *imshowpair* in Matlab to show the difference between the main view-ref view1 and main view-ref view2 before and after the warp. From the two results, Reflection A1, B1, and C1 in the fist row and Reflection A2 and B2 in the second row are all not registered after warping, while our method can still successfully extract the REC pattern as shown in the third column of Figure S3.

## B  Transmission Feature Encoder

The layout of the transmission encoder is depicted in S3. We analyze the necessity of using a hierarchical feature through an ablation study on an example from the RFFR dataset [3], as shown in S4.

## C  Network architecture

The details about the transmission MLP (T-MLP) and the reflection MLP (R-MLP) in our network are depicted in Figure S5, where $\gamma(\cdot)$ refers to the positional encoding function proposed in [4] as

$$\gamma(p) = \left(\sin(2^0 \pi p), \cos(2^0 \pi p), \ldots, \sin(2^{L-1} \pi p), \cos(2^{L-1} \pi p)\right). \tag{S4}$$

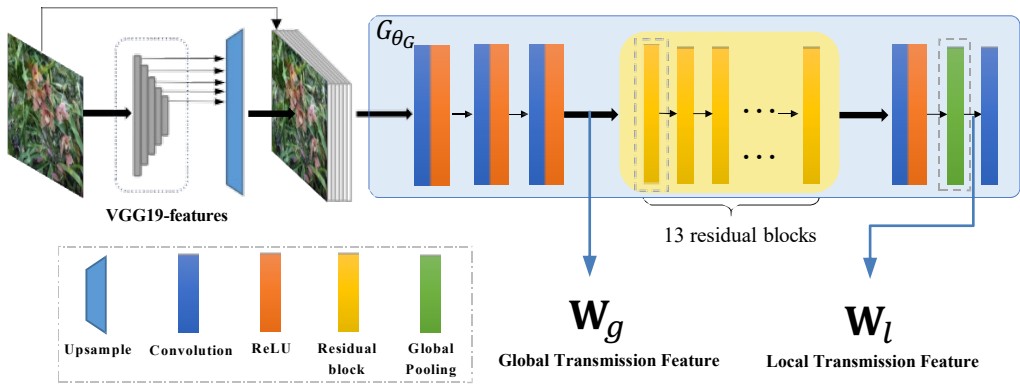

**VGG19-features**

Upsample  Convolution  ReLU  Residual block  Global Pooling

$G_{\theta_G}$

13 residual blocks

$\mathbf{W}_g$
**Global Transmission Feature**

$\mathbf{W}_l$
**Local Transmission Feature**

Figure S3: The structure of the transmission feature encoder and the position of $\mathbf{W}_g$ and $\mathbf{W}_l$.

(a) Input      (b) Our full model      (c) without $\mathbf{W}_g$      (d) Without $\mathbf{W}_l$

Figure S4: Ablation Study on the Multi-scale Transmission Feature Encoder. Please use **Adobe Acrobat** or **KDE Okular** to see the GIF images.

The function $\gamma(\cdot)$ is applied separately to each element of the input coordinate $\mathbf{x}$ and viewing direction $\mathbf{d}$ in $\gamma(\mathbf{x})$ and $\gamma(\mathbf{d})$, respectively. $\mathbf{W}_g$ and $\mathbf{W}_l$ are the global and local features of the images that are projected to the target view, respectively. These features are extracted from the bottleneck layer and the second last layer of a reflection removal network. $\ell_r$ is a learnable reflection embedding for a given view, which models the variational reflection often observed in real-world scenes.

Each blue box in Figure S5 stands for a fully-connected layer, with its number of output channels labeled inside. Our network is divided into a MLP to render transmission (T-MLP) $f_{\mathbf{T}}$, another MLP for reflection (R-MLP) $f_{\mathbf{R}}$ and a branch from T-MLP, $f_\alpha$, to predict the weighting coefficients

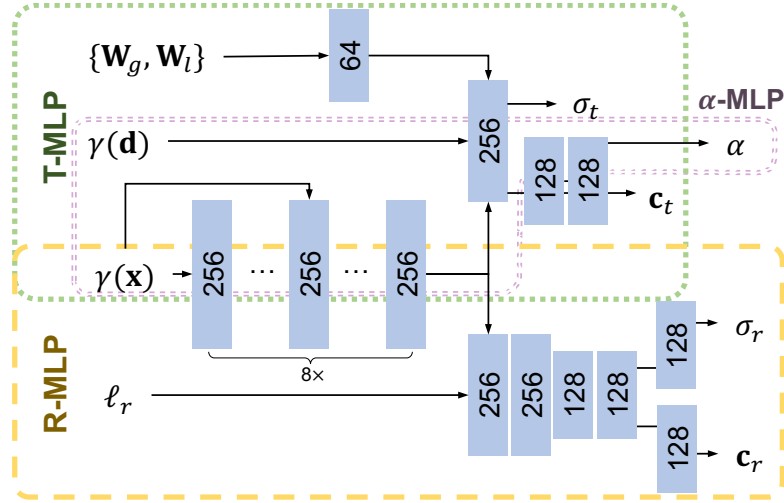

Figure S5: The network architecture of our MLP network.

between reflection and transmission layer. Formally, for the $i$-th target view,

$$f_{\mathbf{T}}(\mathbf{x}, \mathbf{d}, \mathbf{W}_g^{(i)}, \mathbf{W}_l^{(i)}) = (\mathbf{c}_{\mathbf{T}}^{(i)}, \sigma_{\mathbf{T}}^{(i)}),$$
$$f_{\mathbf{R}}(\mathbf{x}, \mathbf{d}, \ell_r^{(i)}) = (\mathbf{c}_{\mathbf{R}}^{(i)}, \sigma_{\mathbf{R}}^{(i)}), \tag{S5}$$
$$f_\alpha(\mathbf{x}, \mathbf{d}) = \alpha.$$

The rendering process is similar to NeRF [4], as we sample a set of points along the ray $\mathbf{r} = \mathbf{o} + t_j \mathbf{d}$,

$$\hat{\mathbf{C}}_S(\mathbf{r}) = \sum_{j=1}^{N} T_{S,j}(1 - \exp{(-\sigma_{S,j}\delta_{S,j})})\mathbf{c}_{S,j}, \text{ where } T_{S,j} = \exp\left(-\sum_{k=1}^{j-1} \sigma_{S,k}\delta_{S,k}\right), \tag{S6}$$

where $S \in \{\mathbf{T}, \mathbf{R}\}$ indicates the scene to be rendered being transmission or reflection. $\delta_i = t_{i+1} - t_i$ is the distance between adjacent sampling points $t_i$ and $t_{i+1}$ along the ray. The weighting coefficients $\alpha$ is used to combine the transmission and reflection as

$$\hat{\mathbf{C}}(\mathbf{r}) = (1 - \alpha)\hat{\mathbf{C}}_{\mathbf{T}}(\mathbf{r}) + \alpha\hat{\mathbf{C}}_{\mathbf{R}}(\mathbf{r}), \tag{S7}$$

where $\hat{\mathbf{C}}(\mathbf{r})$ denotes the rendered color of the ray $\mathbf{r}$.

## D  Setting clarification

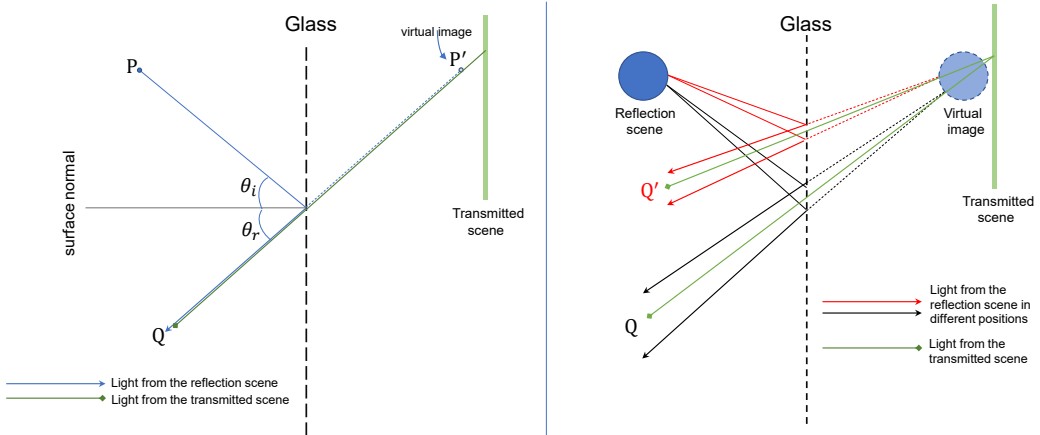

Figure S6: The clarification for two settings. Our setting belongs to the right one.

**Left** of Figure S6: We assume it is caused by some small-scale bright object $\mathbf{P}$ like bulbs or even laser pointers, only occupying limited areas. In this case, its virtual image $\mathbf{P}'$ may indeed be difficult to be observed from different positions.

**Right** of Figure S6: Reflection removal usually deals with larger scenes (we illustrate it using a big circle), which could be observed from different positions ($\mathbf{Q}$ and $\mathbf{Q}'$). When the camera moves from $\mathbf{Q}$ to $\mathbf{Q}'$, we can observe the reflection from different angles (by receiving reflected light rays from different positions), and these reflections are with different appearances. Such differences (also the cues multi-image reflection removal methods rely on) facilitate the separation and rendering in our problem. High-light reflections, as long as they are not too small (like the extreme case of a laser pointer), can also be observed from different angles like the above settings.

## E  More Results

In this section, we show more comparisons of animated results obtained by NeRF [4] and NeRF-W [5]. Note that the figures contain GIF animations that could be displayed properly when viewed with **Adobe Acrobat** or **KDE Okular**. The readers can also find the animated results in the attachment of the supplementary material.

In contrast to the experiments in the main paper, we show more results obtained under 18 training views. From the results shown in Figure S7 and Figure S8, our method can successfully estimate

novel views even only with 6 views for training, while NeRF [4] and NeRF-W [5] can only generate degraded results. From Figure S7, when more training views are provided, NeRF can indeed generate results with higher fidelity, while the undesired reflection is also finally rendered.

(a) Ground Truth (w/o reflection)

(b) Ours, 6 training views

(c) NeRF, 6 training views

(d) NeRF, 12 training views

(e) NeRF, 18 training views

(f) NeRF-W, 18 training views

Figure S7: Results of the scene "sofa" on NeRF [4], NeRF-W [5] and the proposed method. Since this scene is captured through detachable glass, the ground truth without reflection can be captured by removing glass.