# OpenReview forum: "Neural Transmitted Radiance Fields"
_NeurIPS.cc/2022/Conference — NeurIPS 2022 Accept_

### Official Review · Reviewer_2Mxb · 2022-07-01

**Rating:** 5
**Confidence:** 4
**Soundness:** 2 fair
**Presentation:** 3 good
**Contribution:** 2 fair

**Summary:**

This paper proposes a novel view synthesis network specially designed for see-through scenarios. This paper introduces a transmission encoder, which separately estimates the transmission amount against the specular highlight's reflection. In addition, this paper introduces a recurring edge constraint to account for the frequency of edges.

**Questions:**

- As mentioned earlier, specular reflection moves in the opposite direction of the transmitted image when the camera moves. It causes structure-from-motion failures that are supposed to give the right camera pose information as input. This paper doesn't clearly mention how the camera pose is estimated from the input image. I might miss it. If then, please let me know where the information is. I'm worried about the impact of the camera pose when producing other methods' results. I would like to hear more in the rebuttal.

**Ethics Review Area:**

["I don’t know"]

**Limitations:**

Limitations are clearly mentioned in the main paper.

**Strengths And Weaknesses:**

[Strengths]
+ The application and approach of the transmissive scenario sound interesting to me. The specular reflection on glass in the see-through scenario has been rarely discussed in the neural rendering field yet. I found that this new research problem is interesting. Existing solutions such as vanilla NeRF seem to fail when there is a specular reflection in input images, while the proposed method works properly.

[Weaknesses]
- Even though the motivation of the proposed method sounds interesting, I'm not fully sure if this paper is completely developed and evaluated to solve the technical challenges. Specular reflection works very differently from transmission. For instance, when the camera motion occurs, the specular reflection and transmitted image move in opposite directions about the depth position of glass surfaces. The proposed model doesn't seem to account for the physical phenomenon. Instead, it just tries to separate the transmission and reflection along the given view vector, which is not physically plausible. This observation should be valid from a specific view angle. If the method accumulates multiple observations in a voxel grid, the accurate separation cannot be achievable by increasing the number of observations. I would like to hear more in the rebuttal.

- The evaluation of this paper is one of the weakest points. Except for the main results shown in the teaser, most results do not include strong specular reflection. According to the proposed formulation of the recurring edge constraint, the proposed method may work properly when there are strong contrast edges in the transmitted image. The main result of the picture frame is the case. In other cases, the results do not include any strong specular reflection. I think the results look very cherry-picking with a very small number of examples. I would like to see more results to validate the performance of the proposed method.

---

> ### Author Response · Authors · 2022-08-02
> **Response for Official Review of Paper1660 by Reviewer 2Mxb-1st part**
>
> > Even though the motivation of the proposed method sounds interesting, I'm not fully sure if this paper is completely developed and evaluated to solve the technical challenges. Specular reflection works very differently from transmission. For instance, when the camera motion occurs, the specular reflection and transmitted image move in opposite directions about the depth position of glass surfaces. The proposed model doesn't seem to account for the physical phenomenon. Instead, it just tries to separate the transmission and reflection along the given view vector, which is not physically plausible. This observation should be valid from a specific view angle. If the method accumulates multiple observations in a voxel grid, the accurate separation cannot be achievable by increasing the number of observations. I would like to hear more in the rebuttal.
>
> **“Specular reflection works very differently from transmission.”**
>
> We agree that specular reflection works very differently from the transmission, and the transmission and specular reflection may move in opposite directions. However, our model does not rely on such a phenomenon or any given specific viewing angles. In our experiments, we do not deliberately choose specific viewing angles as the main view or reference views.
>
> **“the observation should be valid from a specific viewing angles”**
>
> We plot two figures on this **[link](https://anonymous.4open.science/r/NeurIPS_1660/README.md)** under **7. Specular Reflection** to show our understanding of this question. If we understood correctly, the situation described by Reviewer 2Mxb can be depicted as Figure A. In Figure A, the reflection is mainly caused by light sources (like lightbulbs, flash, or extreme cases like a laser pointer) that occupy limited areas but with very strong intensity causing saturation. When the viewing angle changes, it becomes difficult or even impossible to observe such specular reflection from another angle (especially when the light source is a small point) due to the law of reflection on a mirror-like surface. However, in our context of reflection removal, the reflection occupies much broader areas, as illustrated in Figure B. Such reflection could be observed from different angles (the law of reflection still describes the mirror-like reflection phenomenon, but the camera can receive rays from other directions). Several multi-image reflection removal methods also adopt such assumptions, where the reflection has different shapes and appearances when viewed from different angles, such as [Liu et al. 2020].
>
> [Liu et al. 2020] Liu Y L, Lai W S, Yang M H, et al. Learning to see through obstructions[C]//Proceedings of the IEEE/CVF Conference on Computer Vision and Pattern Recognition. 2020: 14215-14224.
>
> ____________
>
> >  The evaluation of this paper is one of the weakest points. Except for the main results shown in the teaser, most results do not include strong specular reflection. According to the proposed formulation of the recurring edge constraint, the proposed method may work properly when there are strong contrast edges in the transmitted image. The main result of the picture frame is the case. In other cases, the results do not include any strong specular reflection. I think the results look very cherry-picking with a very small number of examples. I would like to see more results to validate the performance of the proposed method.
>
> Our method is not solely designed for strong specular reflection. Thus, our dataset contains both strong and moderate reflections. To further verify this, we show an example by applying our method to the public available NeRF-style dataset, RFFR [Guo et al. 2022], and the results can be found from the **[link](https://anonymous.4open.science/r/NeurIPS_1660/README.md)** under **8. Results for Examples with Specular Reflection**. The stronger specular reflection in this place only occupies limited areas and can also be observed from different positions. We hope it can partly simulate the scenario suggested by the reviewer. Our method can still successfully suppress the strong specular reflection from the results.
>
> [Guo et al. 2022] Guo Y C, Kang D, Bao L, et al. Nerfren: Neural radiance fields with reflections[C]//Proceedings of the IEEE/CVF Conference on Computer Vision and Pattern Recognition. 2022: 18409-18418.

---

> ### Author Response · Authors · 2022-08-02
> **Response for Official Review of Paper1660 by Reviewer 2Mxb-2nd part**
>
> >  Questions: As mentioned earlier, specular reflection moves in the opposite direction of the transmitted image when the camera moves. It causes structure-from-motion failures that are supposed to give the right camera pose information as input. This paper doesn't clearly mention how the camera pose is estimated from the input image. I might miss it. If then, please let me know where the information is. I'm worried about the impact of the camera pose when producing other methods' results. I would like to hear more in the rebuttal.
>
> We apologize for the unclarity. We follow other mainstream NeRF-style methods [NeRF 2020] to estimate the pose using COLMAP. From our experience, COLMAP can accurately extract the camera poses for our tested cases. Even for the strong reflection, if the saturation only occupies limited areas, COLMAP can still accurately extract the transmission poses used for the computation. However, if the strong reflection occupies large areas, COLMAP cannot accurately differentiate the transmission and the reflection. In this situation, it may falsely extract the reflection features to obtain the poses, which affects the subsequent computation.
>
> Since the pose estimation is necessary before any NeRF-style computation, this may influence subsequent rendering. We will include the discussion in our final version.
>
> [NeRF 2020] Mildenhall B, Srinivasan P P, Tancik M, et al. Nerf: Representing scenes as neural radiance fields for view synthesis[C]//European conference on computer vision. Springer, Cham, 2020: 405-421.

---

> ### Author Response · Authors · 2022-08-09
> **An alternative anonymous DropBox link for Reviewer 2Mxb**
>
> **Dear Reviewer 2Mxb , since the external link service provider may have some problems, if you cannot open the **[link](https://anonymous.4open.science/r/NeurIPS_1660/README.md)** in our response, you can find the related documents from this alternative dropbox link:**
>
> https://www.dropbox.com/s/4zx30m8hahcz8cs/Neural%20Transmitted%20Radiance%20Fields%20supplementary.zip?dl=0
>
> **Please download the whole folders and open ‘README.html’. Then, you can find the additional examples we provide for the rebuttal.**

---

### Official Review · Reviewer_Pr2K · 2022-07-06

**Rating:** 5
**Confidence:** 4
**Soundness:** 3 good
**Presentation:** 3 good
**Contribution:** 3 good

**Summary:**

This paper proposes a novel neural radiance field rendering method that is dealing with specular reflection on the object’s surface. The proposed method aims at recovering only the transmission radiance behind the reflection. To that end, this paper proposes to prepare two dedicated networks, i.e. T-MLP and R-MLP, to learn the transmission features and reflection features. This is achieved by applying a single image reflection removal method to the training data to separate the background and the reflection. The learned transmission and reflection color radiance are then combined in a convex combination. In addition, in order to guide the learning of background high-frequency details, this method also applies recurring edge constraints which utilize the observation that background edges appear consistently in multiple different views.

**Questions:**

1. The transmission features W_g and W_i are in different size. How do they feed into the T-MLP network?


**Limitations:**

The limitation of the proposed method is to apply it to any normal scenes or NeRF datasets. If it cannot perform well on non-reflective scenes, the generalizability of the method will be the biggest limitation.

**Strengths And Weaknesses:**

Strengths
1. This paper is generally well-written with clear motivation in the introduction section. It clearly defines the current problem and challenge left by existing NeRF-based methods, which is the reconstruction of scenes behind the transparent surfaces with specular reflection.
2. The comprehensive experiments show that the proposed method consistently outperforms the state-of-the-art methods by a considerable margin, in both qualitative and quantitative evaluations.
3. This paper proposes a new NeRF purpose dataset, which is particularly focusing on the scenes behind the specular reflection. The proposed dataset may impose a strong impact on future research in this area.

Weakness
1. I found the performance comparison with respect to baseline method MVSNeRF is a bit unfair because the selected baseline methods are not designed to deal with reflection, and hence it tends to predict the reflected scene as is. Therefore, the quantitative PSNR results are much worse than the proposed method as expected. Especially, in Figure 3, MVSNeRF almost reconstructs the exact appearance of the target view.
2. For a NeRF method, it is also important to know the performance of the proposed method applied to normal (non-reflective) scenes. Otherwise, the usage of the proposed method is just limited to reflective scenes.  In the submitted paper and supplementary material, all the examples and benchmark data are performed on the scenes with reflection. The authors are suggested to provide more comparison (quantitative) and real normal scene examples in the rebuttal period.
3. What is the processing speed and the network complexity of the proposed method compared to baseline methods? In order to prove the effectiveness of the proposed method, it is crucial to verify that the performance gain is not coming from the extra number of parameters in the network as well as the pre-processed edge map and reflection purged features.
4. From the ablation study, the recurring edge constraints (REC) only bring in very little improvement, but it is considered as one of the two contributions in the method section. It seems that the proposed method is not very effective.
5. It is true that the proposed method outperforms other baselines on the reflective NeRF dataset by a large number. However, the method itself is quite straightforward with limited novelty. It is critical to understand the effectiveness of the proposed method by providing the performance comparison on normal datasets, and hence prove the validity of the proposed method.

---

> ### Author Response · Authors · 2022-08-02
> **Response for Official Review of Paper1660 by Reviewer Pr2K**
>
> >  I found the performance comparison with respect to baseline method MVSNeRF is a bit unfair because the selected baseline methods are not designed to deal with reflection, and hence it tends to predict the reflected scene as is. Therefore, the quantitative PSNR results are much worse than the proposed method as expected. Especially, in Figure 3, MVSNeRF almost reconstructs the exact appearance of the target view.
>
> We agree that MVSNeRF is not designed for issues related to reflection removal. We will consider adding another baseline method, "RR+MVSNeRF", in the final version, for a fair comparison. Through the comparison, it will also be more evident that this setting cannot be directly applied to reconstruct the scenes even if reflection removal has been applied, where constraints of being photometrically static may not hold. We have included this comparison in this **[link](https://anonymous.4open.science/r/NeurIPS_1660/README.md)** under **5. RR+MVS**. Since the reflection removal cannot suppress all reflections, the results obtained by MVSNeRF can also observe reflection residuals.
>
> ____________
>
> >  For a NeRF method, it is also important to know the performance of the proposed method applied to normal (non-reflective) scenes. Otherwise, the usage of the proposed method is just limited to reflective scenes. In the submitted paper and supplementary material, all the examples and benchmark data are performed on the scenes with reflection. The authors are suggested to provide more comparison (quantitative) and real normal scene examples in the rebuttal period.
>
> Thanks for this suggestion. Our method can also achieve robust results under non-reflective scenes. In this situation, the transmission feature extractor can be regarded as a special feature extractor, and REC can be regarded as a module to obtain the edges or gradient of the main view. We conduct more experiments on the LLFF dataset to address this concern. Only 6 views are used for training, and other experiment settings are the same as described in our paper. The results can be found on the **[link](https://anonymous.4open.science/r/NeurIPS_1660/README.md)** under **6. Non-Reflective**. Our method can work properly on non-reflective scenes with sparse views, which further validates the robustness of the proposed framework under the suggested settings.
>
> ____________
>
> >  What is the processing speed and the network complexity of the proposed method compared to baseline methods? In order to prove the effectiveness of the proposed method, it is crucial to verify that the performance gain is not coming from the extra number of parameters in the network as well as the pre-processed edge map and reflection purged features.
>
> Thanks for this suggestion. When comparing the proposed methods with various baselines, we try to control the number of trainable parameters roughly the same. However, we still find that processing speed varies among different methods. A preliminary profiling test shows that the proposed method uses 1.3x the time taken by NeRF-W every epoch and 1.4x the time taken by NeRF. We attribute this inefficiency to the time-consuming homographic warping for high-dimensional transmission features. The computation time might be reduced if we further optimize our implementation, and we will try this before releasing the code.
>
> ____________
>
> >  From the ablation study, the recurring edge constraints (REC) only bring in very little improvement, but it is considered as one of the two contributions in the method section. It seems that the proposed method is not very effective.
>
> Thanks for this suggestion. We also feel interested in the role of REC in the whole framework and have already done some further experiments to validate the effectiveness of REC. One experiment suggested by Reviewer 9GAV can be found in this **[link](https://anonymous.4open.science/r/NeurIPS_1660/README.md)** under **1. REC**, where the results with REC can filter out some reflection residuals than the version without REC.
> ____________
>
> >  It is true that the proposed method outperforms other baselines on the reflective NeRF dataset by a large number. However, the method itself is quite straightforward with limited novelty. It is critical to understand the effectiveness of the proposed method by providing the performance comparison on normal datasets, and hence prove the validity of the proposed method. Normal data -> no reflection data
>
> We have conducted more experiments to verify the effectiveness of the proposed method using more general datasets. The results can be found from the **[link](https://anonymous.4open.science/r/NeurIPS_1660/README.md)** under **6. Non-Reflective**.
> ____________
>
> > Questions: The transmission features W_g and W_i are in different size. How do they feed into the T-MLP network?
>
> In our experiments, they are upsampled using Nearest-Neighbor interpolation to the size of the image.

---

> ### Author Response · Authors · 2022-08-09
> **An alternative anonymous DropBox link for Reviewer Pr2k**
>
> **Dear Reviewer Pr2K , since the external link service provider may have some problems, if you cannot open the **[link](https://anonymous.4open.science/r/NeurIPS_1660/README.md)** in our previous response, you can find the related documents from this alternative dropbox link:**
>
> https://www.dropbox.com/s/4zx30m8hahcz8cs/Neural%20Transmitted%20Radiance%20Fields%20supplementary.zip?dl=0
>
> **Please download the whole folders and open ‘README.html’. Then, you can find the additional examples we provide for the rebuttal.**

---

### Official Review · Reviewer_K1pa · 2022-07-07

**Rating:** 7
**Confidence:** 4
**Soundness:** 3 good
**Presentation:** 3 good
**Contribution:** 3 good

**Summary:**

This paper targets to solve the novel-view synthesis problem with reflection removal, that is, novel-view synthesis of a transmitted object from images corrupted by reflections. A naive baseline, that applies reflection removal techniques to each input image before NeRF, does not solve the problem as the resultant image would not be multi-view consistent; This is because most reflection-removal techniques cannot take advantage of multiple viewpoints. This paper solves this problem by introducing 1) transmission feature integration and 2) recurring edge constraints. First, Transmission feature integration is based on the idea of pixel-NeRF that the feature from other viewpoints can assist the training, and the paper used “transmission feature” instead of the vanilla pixel feature in pixel-NeRF. Second, recurring edge constraints are based on the assumption that a reflected component is sparse in its presence in the aligned image. The paper also collected a new dataset for real multi-view images corrupted by reflections, and the proposed method shows promising results.

**Questions:**

Additional experiments that may further demonstrate the robustness of the proposed method:

- Non-planar reflector. How would the proposed method work when the reflector is not planar? It seems that all the experiments are done with a planar reflector. Does any data include a non-planar reflector? When the reflector is not planar, the behavior of reflections in multi-view images will be very different as the reflected parts will not be aligned after warping. To be specific, when the reflector is planar, the reflected object is equivalent to the virtual object that is behind the reflector and thereby the reflected parts from different viewpoints will be located at the same pixels after warping. However, we cannot expect this alignment in the non-planar reflector as it will be projected differently depending on the viewpoint. This will affect the performance of REC that relies on the aligned edge after warping, and it would be interesting to see how the method works in the case of a non-planar reflector.
- Large reflector. What if the reflector is large enough and thereby the reflections exist in every viewpoint? (e.g., large window as in [9]). This case breaks the assumption used for REC that the reflection is sparse across the different viewpoints, and only “motion” inconsistency can disambiguate reflections and transmissions (though it is still ambiguous to determine which one is transmission).
- Ablation study of feature pyramid W_g and W_l. What if the transmission feature W is used without a pyramid?
- How is the threshold 0.6 in (12) determined?

**Limitations:**

The questions in the above section (Questions) include some limitations that are not handled in the paper: non-planar reflector and large reflector. The proposed method may not work for those cases of reflectors.

**Strengths And Weaknesses:**

### Strengths
- Promising results. The proposed method shows promising results on real multi-view images corrupted by reflections. The comparison with other methods such as NeRF, NeRF-W, and RR + NeRF, also shows that the proposed method performs superior both qualitatively and quantitatively, especially when the number of input images is limited.
- New dataset of multi-view images with and without reflections. The paper shows the newly collected multi-view images, which can facilitate further research on multi-view reconstruction and reflection removal.

### Weaknesses
While the paper proposes an interesting method with promising results, there are some weaknesses that can be improved:
- The presentation of the manuscript can be improved. There are some ambiguous definitions or explanations:
    - [Line 123] What is transmission and reflection entanglement? If it means transmission and reflection have an inherent ambiguity, then the proposed method cannot disambiguate either. “Due to the absorption, reflection, and refractive effect ~” should be further clarified.
    - [Motion inconsistency] The terminology “motion inconsistency” (used frequently all around the paper including the abstract) used for recurring edge constraints is somewhat misleading. The key idea used for recurring edge constraints is that the reflected component may not exist in some viewpoints and thereby have a sparse presence. The reason for this phenomenon is the size of the reflector is limited, which causes the reflected object to be outside the reflector and disappear in some viewpoints. It has nothing to do with motion and thus the term “motion inconsistency” is not the appropriate term to describe the method. Maybe the reflected object is at a different depth from the transmitted object and moves differently in the image (e.g., larger disparity when it is further), but it is not the information that the proposed method directly uses. The description in the main paragraph (line 187-) is already clear, so just choosing a better terminology would improve the clarity of the proposed method.
    - [Line 210] What is \Psi? The notation seems to be not defined.
- Some important details about the transmission feature are missing. What network is used for feature W? From Line 155, I assume the network is based on ERRNet but it is difficult to see which part of the ERRNet is used as there are many components in the ERRNet. Line 162 is not enough for understanding the exact structure. Also, Line 232 explains the pretraining of the transmission encoder briefly and it is somewhat confusing if the method is different from the original ERRNet. The network structure and the training detail needs to be added to the supplemental material.
- Missing baseline. A baseline (that might be interesting) is missing, that is RR + pixel-NeRF (without transmission feature). One of the main contributions of this paper is using the transmission feature, which is the combination of 1) reflection removal and 2) pixel-NeRF (assist the training of NeRF). If these two parts are divided into the reflection removal part and the pixel-NeRF part, it can be another baseline of RR + pixel-NeRF, which will be a more fair and interesting baseline.
- REC has a limited performance (at least quantitatively). The second main contribution of this paper is using recurring edge constraints (REC), but the effect of REC seems to be marginal quantitatively as shown in the ablation study (Table 2). The PSNR  without REC is 22.48, which is almost the same as that of the complete model (22.75). It would be interesting to see how REC works in more challenging data.

---

> ### Author Response · Authors · 2022-08-02
> **Response for Official Review of Paper1660 by Reviewer K1pa-1st part**
>
> >  The presentation of the manuscript can be improved. There are some ambiguous definitions or explanations:[Line 123] What is transmission and reflection entanglement? If it means transmission and reflection have an inherent ambiguity, then the proposed method cannot disambiguate either. “Due to the absorption, reflection, and refractive effect ~” should be further clarified.
>
> **“What is transmission and reflection entanglement?”**
>
> By "transmission and reflection entanglement", we mean that the accurate separation of the transmission $\mathbf{B}$ and the reflection $\mathbf{R}$ is an ill-posed problem, which is recognized in reflection-removal-related areas. We agree that our method cannot "disambiguate" them, while we hope to make their separation as reasonable as possible under the current framework. We also realize that the "entanglement" and "disambiguate" in this place are not clear and accurate enough. In the final version, we will clearly say that their separation is an ill-posed problem, and our goal is to provide a reasonable separation under the current framework.
>
> **“Due to the absorption, reflection, and refractive effect ~ should be further clarified.”**
>
> The absorption, reflection, and refractive effects denote several factors that may influence the light emitted by the objects on both sides of the glass. When light travels through a piece of glass, the light's intensity is typically influenced by the absorption and reflectivity effect [Wan et al. 2022]. The refractive effect is related to the density of glass and mainly affects the relationship between the transmission and the reflection. These factors jointly make reflection separation a difficult task. More details about the three factors can be found in this paper [Wan et al. 2022]. This part will be further clarified in the final version.
>
> [Wan et al. 2022] Wan R, Shi B, Li H, et al. Benchmarking single-image reflection removal algorithms[J]. IEEE Transactions on Pattern Analysis and Machine Intelligence, 2022.
>
> ______
>
> > [Motion inconsistency] The terminology “motion inconsistency” (used frequently all around the paper including the abstract) used for recurring edge constraints is somewhat misleading. The key idea used for recurring edge constraints is that the reflected component may not exist in some viewpoints and thereby have a sparse presence. The reason for this phenomenon is the size of the reflector is limited, which causes the reflected object to be outside the reflector and disappear in some viewpoints. It has nothing to do with motion and thus the term “motion inconsistency” is not the appropriate term to describe the method. Maybe the reflected object is at a different depth from the transmitted object and moves differently in the image (e.g., larger disparity when it is further), but it is not the information that the proposed method directly uses. The description in the main paragraph (line 187-) is already clear, so just choosing a better terminology would improve the clarity of the proposed method.
>
> Thanks for this helpful suggestion. It provides a better perspective to consider an important component used in our framework. We agree that this phenomenon is due to the limited size of the reflector, and it causes the reflected object to be outside the viewpoints. After carefully considering reviewers' suggestions, we will directly use "Recurring Edge Constraint" in the final version.
> ____________
> > [Line 210] What is \Psi? The notation seems to be not defined.
>
> Eq. (14) below line 210 have two lines, and the definition for $\Psi$ has already been given at the second line. It defines a pixel-wise correlation between the transmission and reflection, which helps to separate them in the gradient domain.
> ____________
> >  Some important details about the transmission feature are missing. What network is used for feature W? From Line 155, I assume the network is based on ERRNet but it is difficult to see which part of the ERRNet is used as there are many components in the ERRNet. Line 162 is not enough for understanding the exact structure. Also, Line 232 explains the pretraining of the transmission encoder briefly and it is somewhat confusing if the method is different from the original ERRNet. The network structure and the training detail needs to be added to the supplemental material.
>
> We apologize for this unclarity. We will clarify this in the final version, and its details can be found in this **[link](https://anonymous.4open.science/r/NeurIPS_1660/README.md)** with **2. ERRNet-illustration**. For the pretraining of this transmission feature extractor, we follow the strategy proposed in ERRNet with its released training data. Thus, the model is approximately equivalent to its released model.

---

> ### Author Response · Authors · 2022-08-02
> **Response for Official Review of Paper1660 by Reviewer K1pa-2nd part**
>
>
> >  Missing baseline. A baseline (that might be interesting) is missing, that is RR + pixel-NeRF (without transmission feature). One of the main contributions of this paper is using the transmission feature, which is the combination of 1) reflection removal and 2) pixel-NeRF (assist the training of NeRF). If these two parts are divided into the reflection removal part and the pixel-NeRF part, it can be another baseline of RR + pixel-NeRF, which will be a more fair and interesting baseline.
>
> We agree that RR+Pixel-NeRF may reflect interesting phenomena and new insights. However, its data loader is designed for the data generated by Blender and needs much effort to adapt to our case. We are afraid we cannot figure this out given the tight time during rebuttal. As an alternative, MVSNeRF is similar to PixelNeRF and uses PixelNeRF as a baseline, which can be found in Table 1 and Table 2 of [Chen et al. 2021]. We instead run a setting as RR+MVSNeRF, and this experiment can be found in this **[link](https://anonymous.4open.science/r/NeurIPS_1660/README.md)** under **5. RR+MVS**. Our method still performs better because it can suppress reflection within the whole framework. We will continue working on the RR+PixelNeRF setup and include this comparison in the final version.
>
> [Chen et al. 2021] Chen A, Xu Z, Zhao F, et al. Mvsnerf: Fast generalizable radiance field reconstruction from multi-view stereo[C]//Proceedings of the IEEE/CVF International Conference on Computer Vision. 2021: 14124-14133.
>
> ____________
>
> >  REC has a limited performance (at least quantitatively). The second main contribution of this paper is using recurring edge constraints (REC), but the effect of REC seems to be marginal quantitatively as shown in the ablation study (Table 2). The PSNR without REC is 22.48, which is almost the same as that of the complete model (22.75). It would be interesting to see how REC works in more challenging data.
>
> We appreciate this suggestion. It is also mentioned by Reviewer 9GAV. We follow the suggestion made by Reviewer 9GAV to show the influence in a "slowly-relaxed" but more challenging setting. Specifically, we achieve this goal by using synthetic images with gradually changing parameters (to mimic moderately and highly reflective surfaces) for reflection components and making the reflection components cover the whole image plane. The results can be found from this **[link](https://anonymous.4open.science/r/NeurIPS_1660/README.md)** under **1. REC**. In the first experiment with $\mathbf{I} = 0.6\mathbf{B}+0.4\mathbf{R}$, for the reflection (at the center of *Our Model without REC*) that are hard to be further suppressed by the transmission feature extractor, REC can further exclude them during the rendering process. However, we agree that REC cannot function well under the slivered-mirror case with $\mathbf{I} = 0.2\mathbf{B}+0.8\mathbf{R}$. In this situation, since COLMAP can only extract dominant reflection features, REC cannot extract the transmission patterns (the flower, which becomes almost invisible to human vision in this case).
> ____________

---

> ### Author Response · Authors · 2022-08-02
> **Response for Official Review of Paper1660 by Reviewer K1pa-3rd part**
>
> >  Non-planar reflector. How would the proposed method work when the reflector is not planar? It seems that all the experiments are done with a planar reflector. Does any data include a non-planar reflector? When the reflector is not planar, the behavior of reflections in multi-view images will be very different as the reflected parts will not be aligned after warping. To be specific, when the reflector is planar, the reflected object is equivalent to the virtual object that is behind the reflector and thereby the reflected parts from different viewpoints will be located at the same pixels after warping. However, we cannot expect this alignment in the non-planar reflector as it will be projected differently depending on the viewpoint. This will affect the performance of REC that relies on the aligned edge after warping, and it would be interesting to see how the method works in the case of a non-planar reflector.
>
> **“Non-planar reflector”**
>
> Thanks for this suggestion. Most reflection-removal-related problems assume a piece of planar glass. We follow this assumption in this paper and do not specifically consider the influence of non-planar reflectors in our experiments. We will clarify this assumption in the final version. Since we capture images in the real world, some examples in Figure 3 of our paper are captured through a piece of glass with slightly curved areas, and our method still shows its robustness.
>
> **“We cannot expect this alignment in the non-planar reflector as it will be projected differently depending on the viewpoint.”**
>
> We agree that the reflection components may not be aligned for non-planar reflectors under some situations, while REC only needs to find the recurring transmission components. Thus, the unaligned reflection components are not a big issue in this place. We show an example in this **[link](https://anonymous.4open.science/r/NeurIPS_1660/README.md)** under **3. Non-planar** to better explain it. In this example, the reflection components are not aligned after the warping, while our method can still identify the corresponding transmission edges and reflection edges.
>
> We try to analyze the influence of the non-planar glass on our framework. If such non-planar glass distorts the light emitted by the transmission scenes behind the glass, it may influence the rendering process. In this situation, extracting the necessary transmission poses may become difficult due to the distorted transmission details.
>
> We agree that the non-planar glass is an interesting problem for NeRF with reflections, which is worth to be explored further. We will carefully consider it in our future study.
> ____________
>
> >  Large reflector. What if the reflector is large enough and thereby the reflections exist in every viewpoint? (e.g., large window as in [9]). This case breaks the assumption used for REC that the reflection is sparse across the different viewpoints, and only “motion” inconsistency can disambiguate reflections and transmissions (though it is still ambiguous to determine which one is transmission).
>
> This is a very good question. The reflection may only dominate limited regions in many situations due to its regional property [Wan et al. 2022]. Thus, the assumption used for REC is still a valid approximation. From our answer to Reviewer 9GAV's question and the results shown in this **[link](https://anonymous.4open.science/r/NeurIPS_1660/README.md)** under **1. REC**, even when the reflection occupies larger areas, our method can still work correctly if COLMAP can extract the transmission pose well. COLMAP fails to accurately estimate the transmission pose needed for the warp in slivered-mirror scenarios (**I**  = 0.2**B**+0.8**R**), where the reflection almost occludes the light rays emitted by the transmission scene. In this situation, transmission REC cannot be extracted.
>
> [Wan et al. 2022] Wan R, Shi B, Li H, et al. Benchmarking single-image reflection removal algorithms[J]. IEEE Transactions on Pattern Analysis and Machine Intelligence, 2022.
>
> ____________
>
> >  Ablation study of feature pyramid W_g and W_l. What if the transmission feature W is used without a pyramid?
>
> We have shown more ablation studies for this design, where the results can be found from the following **[link](https://anonymous.4open.science/r/NeurIPS_1660/README.md)** under **4. Ablation**. The complete model can output more robust results than the model without $\mathbf{W}_g$ and $\mathbf{W}_l$, though the two incomplete models can all suppress the reflection to some degrees.
>
> ____________
> >  How is the threshold 0.6 in (12) determined?
>
> This threshold is determined empirically. This threshold can filter out some small gradient values belonging to the reflection components. We search from 0 to 1 with a step of 0.1 and fix it as 0.6 in our experiments.

---

> ### Author Response · Authors · 2022-08-09
> **An alternative anonymous DropBox link for Reviewer K1pa**
>
> **Dear Reviewer K1pa, since the external link service provider may have some problems, if you cannot open the **[link](https://anonymous.4open.science/r/NeurIPS_1660/README.md)** in our response, you can find the related documents from this alternative dropbox link:**
>
> https://www.dropbox.com/s/4zx30m8hahcz8cs/Neural%20Transmitted%20Radiance%20Fields%20supplementary.zip?dl=0
>
> **Please download the whole folders and open ‘README.html’. Then, you can find the additional examples we provide for the rebuttal.**

---

> ### Comment · Reviewer_K1pa · 2022-08-09
> **Response to authors**
>
> Dear authors,
>
> Thank you for uploading an updated version of the manuscript and an HTML file. The HTML file was really helpful in understanding how each concern from reviewers is addressed.
>
> Most of my concerns are clearly addressed, including better terminology and explanations about reflection (i.e., reflection entanglement and motion inconsistency), details for reproduction (i.e., ERRNet-illustration), and additional experiments (i.e., RR + MVS, REC, non-planar reflectors, large reflectors). In particular, the additional experiments improved the clarity of the proposed method a lot by showing which component is critical for the performance (e.g., RR + MVS), and some challenging situations that the proposed method can still handle (e.g., non-planar reflectors and large reflectors). I became more positive about this paper.

---

> ### Author Response · Authors · 2022-08-10
> **Thanks for your response**
>
> Dear Reviewer K1pa, thanks for you kind reply very much. We are glad to have this opportunity to address your concerns. We will continue improving our paper to make it better.

---

### Official Review · Reviewer_9GAV · 2022-07-10

**Rating:** 7
**Confidence:** 3
**Soundness:** 3 good
**Presentation:** 3 good
**Contribution:** 3 good

**Summary:**

The paper proposes a method to learn neural radiance fields that represent the underlying scene free of reflective components in the scene, i.e. explicitly represented the transmitted regions of the scene. Prior work in representing transmitted radiance field relies on reflection removal from the input image sequence, however this is a challenging problem and typically results in photometric inconsistencies. The proposed method uses a novel formulation leveraged on the observation that reflective components in the radiance field are sparser than the transmitted components. A patch-based rendering scheme is used to handle the local characteristics of reflective/transmissive components.

**Questions:**

It would be an interesting study to see the performance of this method as the sparsity assumption is slowly relaxed, where the reflective components in the scene change from transmissive glass plates on one end of the spectrum to fully silvered mirrors on the other. Have the authors performed any experiments on such scenes?

While occlusions are one of the main limiting factors, occlusions in the reflected part of the scene however are not likely to cause a problem in my opinion, since the edge constraint assumption and the sparsity assumption are still valid?

**Limitations:**

Yes, the authors discuss the limitations of the work

**Strengths And Weaknesses:**

Strengths:
The paper is well written and the exposition is clear. The paper provides a through introduction and a motivation for the solution, before properly explaining the proposed solution. As such I find the paper to be a useful contribution to the community and beneficial for the reader.
The use of the transmission encoder with pyramid-scale features is interesting and the choice of Wg and Wl is properly motivated.
The recurring edge constraints are the core strength of the paper and the description provided in section 4.2 is succinct.
The qualitative and quantitative results in the paper and supplemental material clearly demonstrates that the transmitted radiance field is captured free form noise due to reflection.

Weaknesses:
The authors rightly point out that weighting coefficients are dependent on several factors. The viewing direction (wrt lights in the scene) and the camera position are correlated and more discussion is warranted on whether an MLP that encodes the weighting coefficients is sufficient in general.

---

> ### Author Response · Authors · 2022-08-02
> **Response for Official Review of Paper1660 by Reviewer 9GAV**
>
> > Weaknesses: The authors rightly point out that weighting coefficients are dependent on several factors. The viewing direction (wrt lights in the scene) and the camera position are correlated and more discussion is warranted on whether an MLP that encodes the weighting coefficients is sufficient in general.
>
> We apologize for the missing details about weighting coefficients $\alpha$, which may have misled the reviewer into doubting the effectiveness of the MLP used here. In the inline equation $f_\alpha(\mathbf{x}, \mathbf{d})$ at line 180, $\mathbf{x}$ refers to the position of any given point, and $\mathbf{d}$ is the viewing direction, consistent with the notation defined at line 106 and line 107. The weighting map of a given view is rendered similarly to Eq.(2), where $\sigma_{t}^{(i)}$ being the $\sigma$  in the equation, and the value of $\alpha$ along the ray emitted from the camera is accumulated. This setting enables the network to increase its robustness with real-world cases in our experiments. These details will be clarified in the final version.
> ___
> >  Question 1: It would be an interesting study to see the performance of this method as the sparsity assumption is slowly relaxed, where the reflective components in the scene change from transmissive glass plates on one end of the spectrum to fully silvered mirrors on the other. Have the authors performed any experiments on such scenes?
>
> This is a very inspiring suggestion. Before we answer this question, we define $\mathbf{I}$, $\mathbf{B}$, and $\mathbf{R}$ as the mixture image captured through glass, the transmission components, and the reflection components. This definition follows the common settings in reflection-removal-related works.
>
> We use $\mathbf{I}  = 0.2\mathbf{B}+0.8\mathbf{R}$ (relatively high weight on reflection) to mimic the ``silvered-mirror" case. The results are given in this **[link](https://anonymous.4open.science/r/NeurIPS_1660/README.md)** under **1. REC**. The COLMAP used for the pose extraction cannot obtain the transmission features, which the reflection has almost totally occluded.
>
> We adjust the weight to mimic the transmissive glass with more common reflectivity with $\mathbf{I} = 0.6\mathbf{B}+0.4\mathbf{R}$. COLMAP can well extract the pose information for REC computation, and our proposed scheme can also work properly.
> ______
> >  Question 2: While occlusions are one of the main limiting factors, occlusions in the reflected part of the scene however are not likely to cause a problem in my opinion, since the edge constraint assumption and the sparsity assumption are still valid?
>
> The edge constraint and sparsity assumptions are still valid even if some of the reflection components are occluded in certain views. Thus, we do not consider the occlusions in the reflected part as a limiting factor.

---

> ### Author Response · Authors · 2022-08-09
> **An alternative anonymous DropBox link for Reviewer 9GAV**
>
> **Dear Reviewer 9GAV, since the external link service provider may have some problems, if you cannot open the **[link](https://anonymous.4open.science/r/NeurIPS_1660/README.md)** in our response, you can find the related documents from this alternative dropbox link:**
>
> https://www.dropbox.com/s/4zx30m8hahcz8cs/Neural%20Transmitted%20Radiance%20Fields%20supplementary.zip?dl=0
>
> **Please download the whole folders and open ‘README.html’. Then, you can find the additional examples we provide for the rebuttal.**

---

### Author Response · Authors · 2022-08-02
**Thanks for the suggestions made by the reviewing panel members.**

We appreciate the comments and efforts made by our reviewing panel members. We hope to use this opportunity to provide more details about our work. Thanks very much.

---

### Author Response · Authors · 2022-08-07
**Status update**

Dear All, we are revising our paper based on the precious suggestions from each reviewer. We will update our paper as soon as possible.

---

### Author Response · Authors · 2022-08-08
**Summary of Changes**

Dear Reviewers,

We appreciate every comment given by our reviewers. If you have any additional questions or concerns, please let us know by the end of this Author-Reviewer Discussion Stage (Aug 9th).

We again thank all reviewers for your time reviewing and helping to improve our paper. We have updated our paper and the supplementary after considering your suggestions. You can find them from the newly submitted files. All revisions to our original paper and supplementary materials are denoted using the **red** font. We will continue to proofread our paper to avoid any typos.

Besides, more animated results are also provided as independent files in the supplementary.

**You can find our original paper in the revision history under the timestamp 17 May 2022**.

For reviewer 9GAV:

1. We have better explained $f_\alpha(\mathbf{x}, \mathbf{d})$ in the revised manuscript. More discussions have been settled to explain its usage here (**line 181-line 184**).
2. Due to the page limitation, we put the discussions for REC into the supplementary materials (**Under Section A**).

For reviewer K1pa:

1. We have replaced the “entanglement” with “the reflection interference.” The word “disambiguate” has been replaced with “separate” or “separation.”
2. We have rephrased the paragraph with "the absorption, reflection, and refractive effect” in the revised manuscript to avoid unclearity (**line 120-line 124**).
3. “Motion inconsistency” has been replaced with “Recurring edge constraints.” Please search **"Recurring edge"** or **"Recurring edge constraints"** in our paper.
4. We have explained $\Psi$ in the revised manuscript (**line 217-line 218**).
5. The pretraining of ERRNet is provided in the revised manuscript (**line 237-line 239**).
6. The way we make use of $\mathbf{W}_g$ and $\mathbf{W}_l$ is also provided in **Section B** of the supplementary.
7. RR+MVSNeRF has been considered as a baseline in the experiments (**Table 1, Figure 3, and Figure 4**). We admit that RR+PixelNeRF will be added to our final version.
8. We have discussed the non-planar glass in the revised manuscript (**line 247-line 248**) and clearly show its challenge in the Limitations (**line 314**).
9. We also discuss the challenge of large reflectors in the Limitations (**line 313 - line 314**).
10. Due to the page limitation, we include the ablation study of $\mathbf{W}_g$ and $\mathbf{W}_l$ in the supplementary materials (**Section B**). At the same time, we have cited related papers as the reference for such a design (**line 161 - line 163**).
11. We have discussed how we set the threshold in the revised manuscript (**line 207 - line 208**).

For reviewer Pr2k

1. We have shown the quantitative values of RR+MVSNeRF in **Table 1** and the visual comparison in **Figure 3 and Figure 4** of the revised manuscript.
2. We have shown two examples with the non-reflective scenes in **Figure 6** of the revised manuscript.
3. Due to the page limitation, more discussions about REC has been put into the supplementary materials (**Under Section A**).
4. We have shown how $\mathbf{W}_g$ and $\mathbf{W}_l$ are used in our work (**line 163 - line 164**).

For reviewer 2Mxb

1. After checking our paper, besides the teaser, we have already included two examples with specular reflection in **Figure 3** of the original manuscript, which shows our method can handle the specular reflection. We further include two more examples with specular reflection in **Figure S4** and **Figure S8** of the revised supplementary material.
2. We have clearly shown that we obtained the pose via the COLMAP (**line 245-line 246**) and also discussed when COLMAP might fail in the Limitation. We finally propose some possible solutions for our future work to be considered (**line 315 - line 316**).
3. Due to the page limitation, we clarify the differences between the settings pointed out by the reviewer (the 1st weakness) and ours in **Section D** of the revised supplementary material.

---

### Meta-Review · Area_Chair_i29N · 2022-08-27

**Recommendation:** Accept
**Confidence:** Certain

**Metareview:**

This paper proposes a novel neural radiance field rendering method that is dealing with specular reflection on the object’s surface. The authors present a novel method to solve the limitation of the existing NeRF-based methods for the scenes behind the transparent surfaces with specular reflection. The review results are two A(7) and two BA(5). After carefully checking out the rebuttals and discussions, I recommend the paper to be accepted for this NeurIPS.

**Award:**

No

---

### Decision · Program_Chairs · 2022-09-14

Accept